

# Reproducible research and GIScience: an evaluation using AGILE conference papers

Daniel Nüst[1], Carlos Granell[2], Barbara Hofer[3], Markus Konkol[1], Frank O. Ostermann[4], Rusne Sileryte[5] and Valentina Cerutti[4]

[1] Institute for Geoinformatics, University of Münster, Münster, Germany
[2] Institute of New Imaging Technologies, Universitat Jaume I de Castellón, Castellón, Spain
[3] Interfaculty Department of Geoinformatics - Z_GIS, University of Salzburg, Salzburg, Austria
[4] Faculty of Geo-Information Science and Earth Observation (ITC), University of Twente, Enschede, The Netherlands
[5] Faculty of Architecture and the Built Environment, Delft University of Technology, Delft, The Netherlands

## ABSTRACT

The demand for reproducible research is on the rise in disciplines concerned with data analysis and computational methods. Therefore, we reviewed current recommendations for reproducible research and translated them into criteria for assessing the reproducibility of articles in the field of geographic information science (GIScience). Using this criteria, we assessed a sample of GIScience studies from the Association of Geographic Information Laboratories in Europe (AGILE) conference series, and we collected feedback about the assessment from the study authors. Results from the author feedback indicate that although authors support the concept of performing reproducible research, the incentives for doing this in practice are too small. Therefore, we propose concrete actions for individual researchers and the GIScience conference series to improve transparency and reproducibility. For example, to support researchers in producing reproducible work, the GIScience conference series could offer awards and paper badges, provide author guidelines for computational research, and publish articles in Open Access formats.

## INTRODUCTION

A "reproducibility crisis" has been observed and discussed in several scientific disciplines such as economics (*Ioannidis, Stanley & Doucouliagos, 2017*), medical chemistry (*Baker, 2017*), neuroscience (*Button et al., 2013*), and for scientific studies in general, across various disciplines (*Ioannidis, 2005*). The lack of reproducibility in scientific studies stems from researchers facing challenges in understanding and re-creating others' results, a situation that is common in data-driven and algorithm-based research. However, even though algorithms are becoming more relevant in GIScience, a reproducibility crisis has not yet been observed in this field. In GIScience, failures to reproduce are not yet a topic of

Corresponding author
Daniel Nüst,
daniel.nuest@uni-muenster.de

broad and common interest, but this field should be working to prevent a crisis instead of reacting to one. Given this motivation, we aim to adapt the observations and challenges of reproducible research from other disciplines to the GIScience community, and then use these adapted criteria to assess the reproducibility of research produced by members of this field and presented at a conference for the Association of Geographic Information Laboratories in Europe (AGILE), which has organised annual conferences on GIScience topics since 1998 (https://agile-online.org/index.php/conference/past-agile-conferences; all links last accessed Nov 23 2017). The conference series's broad topical scope and its notoriety in the GIScience community make it a reasonable starting point to investigate the level of reproducibility in GIScience research. This publication continues a collaboration started at the AGILE 2017 pre-conference workshop "Reproducible Geosciences Discussion Forum" (http://o2r.info/reproducible-agile/2017/).

In this work, we first review papers from other disciplines, which provide recommendations on how to make research more transparent and reproducible. This literature study provides the general criteria we used to systematically evaluate a sample of 32 AGILE conference papers from the last eight years. From this evaluation and the lessons learned by others, we formulate recommendations for the AGILE community, ranging from individual researchers' practises to practises to be carried out by conference organisations. Because of its international reach, broad range of topics, and long-sustained community, we argue that AGILE is in a unique position to take a leading role to promote reproducibility in GIScience. The following research questions (RQs) structure the remainder of this article:

RQ 1 *What are general criteria for reproducible research?*
RQ 2 *What are key criteria for reproducible research in GIScience?*
RQ 3 *How do AGILE conference papers meet these reproducibility criteria?*
RQ 4 *What strategies could improve reproducibility in AGILE contributions and GIScience in general?*

'Related work' provides references targeting RQ 1, which are detailed further in 'Assessment of Reproducibility' to address RQ 2. The results of applying the criteria ('Results') answer RQ 3, and the discussion ('Discussion') responds to RQ 4.

## RELATED WORK

Reproducible research is a frequently discussed topic in editorials and opinion articles in high-impact journals (cf. 'Recommendations and suggestions in literature'). Extensive studies on the state of reproducibility have been conducted in some domains, e.g., in computer systems research (*Collberg & Proebsting, 2016*, see also project website http://reproducibility.cs.arizona.edu/) or bioinformatics (*Hothorn & Leisch, 2011*). For the field of geoscience research, some discussion of reproducibility has happened sporadically for quantitative geography (*Brunsdon, 2016*), cartography (*Giraud & Lambert, 2017*) and volunteered geographic information (VGI) (*Ostermann & Granell, 2017*), but no comprehensive study of reproducibility in the GIScience domain has been conducted.

Even though recent studies highlight an increased awareness of and willingness for open research, they also draw attention to persistent issues and perceived risks

associated with data sharing and publication, such as the lack of rewards and the risk of losing recognition in a competitive academic environment (*Tenopir et al., 2011*; *Ioannidis, 2014*). Beyond individual concerns, there are systematic impediments. Some studies have mentioned that reproducible research is not in the individual researcher's domain but instead is a multi-actor endeavour, which requires a collective mind shift within the scientific community (*Stodden et al., 2016*; *McNutt, 2014*; *Ioannidis, 2014*). Funding agencies, research institutions, publishers, journals, and conferences are all responsible for promoting reproducible research practises. Existing examples (journals welcoming reproducible papers: *Information Systems* (https://www.elsevier.com/journals/information-systems/0306-4379), *Vadose Zone Journal* (https://dl.sciencesocieties.org/publications/vzj/articles/14/10/vzj2015.06.0088), *GigaScience* (https://academic.oup.com/gigascience/pages/instructions_to_authors), *JASA* (http://www.sph.umn.edu/news/wolfson-named-reproducibility-editor-asa-statistics-journal/)) are remarkable, yet in general they are scarce and testimonial.

Another hindrance to reproducible research is that, given the distinct nature and variety of research practises, the term reproducibility has been used with varying meanings and may stand for repeatability, robustness, reliability or generalisability of scientific results (*Editorial, 2016*). There has been some confusion about contradictory meanings in the literature (see for example Mark Liberman's "Replicability vs. reproducibility" (http://languagelog.ldc.upenn.edu/nll/?p=21956)). Wikipedia's definition (https://en.wikipedia.org/wiki/Reproducibility) is widely used to distinguish both terms:

> Reproducibility is the ability to get the same research results using the raw data and computer programs provided by the researchers. A related concept is replicability, meaning the ability to independently achieve similar conclusions when differences in sampling, research procedures and data analysis methods may exist.

*Leek & Peng (2015)* similarly define reproducibility as the ability to compute exactly the same results of a study based on original input data and details of the analysis workflow. They refer to replicability as obtaining similar conclusions about a research question derived from an independent study or experiment. A *Nature Editorial (2016)* defines reproducibility as achieved when "another scientist using the same methods gets similar results and can draw the same conclusions". *Stodden et al.* (*2016*, p. 1240) base their reproducibility enhancement principles on "the ability to rerun the same computational steps on the same data the original authors used". While most statements in the literature show that researchers have a common understanding of what these two concepts mean, the interpretation and application of these concepts by the scientific communities is still inconsistent and leads to different methods and conventions for disseminating scientific work. In the field of GIScience, *Ostermann & Granell* (*2017*, p. 226) argue that "a reproduction is always an exact copy or duplicate, with exactly the same features and scale, while a replication resembles the original but allows for variations in scale for example". Hence, reproducibility is exact whereas replicability means confirming the original conclusions, though not necessarily with the same input data, methods, or results.
## MATERIALS & METHODS

### The paper corpus

We consider the AGILE conference series publications to be a representative sample of GIScience research because of the conferences' broad topical scope. Since 2007, the AGILE conference has had a full paper track (cf. *Pundt & Toppen, 2017*) and a short paper track with blind peer review. The latter is published for free on the AGILE website. Legal issues (full paper copyrights lie with the publisher Springer, see https://agile-online.org/index.php/conference/springer-series) and practical considerations (assessment of reproducibility is a manual time-consuming process; old publications introduce bias because of software unavailability) led us to choose to apply our evaluation only to nominees for the "best full and short paper" awards for 2010, and 2012 to 2017 (no records for a best paper award could be found for 2011). Typically, there are three full paper and two short paper candidates per year (https://agile-online.org/index.php/conference/proceedings). Exceptions are 2013 with only two full papers and 2010 without any short papers. The corpus contains 32 documents: 20 full papers (7.9% of 253 full papers since 2007) and 12 short papers[1].

An exploratory text analysis of the paper corpus investigated the occurrence of keywords related to reproducibility, data, and software. The code is published as an executable document[2] (cf. *Nüst, 2018*). Most frequent terms mentioned are illustrated by Fig. 1. Table 1 shows keyword occurrence per paper and in the entire corpus (bottom row "Total"). Keyword identification uses word stems, e.g., `reproduc` includes "reproducible", "reproduce", and "reproduction" (see *Nüst (2018)* for details). While this matches common and established (technical) terms, it might not capture all phrases an author could use to describe reproducibility-related aspects of the work. Putting these corner cases aside, the numbers are clear enough to draw the following conclusions. Few papers mention reproducibility, some mention code and software, and many mention processes, algorithms, and data. This points to data and analysis being generally discussed in the publications, while being able to recreate the data and analyses is not deliberated.

### Assessment of reproducibility
#### Recommendations and suggestions in literature

Scientists from various disciplines suggest guidelines for open and reproducible research considering the specific characteristics of their field, e.g., *Sandve et al. (2013)* for life sciences, *McNutt (2014)* for field sciences, and *Gil et al. (2016)* for the geoscience paper of the future. Our goal was to first identify common recommendations that are applicable across research fields, including GIScience.

Suggested guidelines found in the reproducibility-related papers we investigated were categorised according to four aspects: *data* concerns all inputs; *methods* cover everything on the analysis of data, e.g., algorithms, parameters, and source code; *results* include (intermediate) data and parameters as well as outcomes such as statistics, maps, figures, or new datasets; and *structure* considers the organisation and integration of the other aspects. While some of the publications focus on specific aspects such as data (*Gewin, 2016*), code (*Stodden & Miguez, 2014*), workflow semantics (*Scheider, Ostermann & Adams, 2017*), and

[1] Full number of short papers cannot be derived automatically, (cf. *Nüst, 2018*).

[2] Using R Markdown, see http://rmarkdown.rstudio.com/.
**A** **B**

| place | word | n | # papers |
|---|---|---|---|
| 1 | data | 1058 | 31 |
| 2 | information | 589 | 32 |
| 3 | spatial | 577 | 30 |
| 4 | map | 411 | 25 |
| 5 | model | 411 | 25 |
| 6 | building | 381 | 24 |
| 7 | time | 378 | 30 |
| 8 | approach | 297 | 32 |
| 9 | osm | 292 | 8 |
| 10 | buildings | 266 | 15 |
| 11 | geographic | 249 | 28 |
| 12 | location | 239 | 26 |
| 13 | analysis | 229 | 28 |
| 14 | users | 225 | 19 |
| 15 | results | 207 | 30 |
| 16 | web | 206 | 21 |
| 17 | models | 202 | 20 |
| 18 | values | 202 | 23 |
| 19 | patterns | 196 | 16 |
| 20 | maps | 189 | 20 |

**Figure 1** Two illustrations of the test corpus papers: word cloud, scaled and coloured by number of occurrence of words with at least 100 occurrences (96 unique words) (A); top words sorted by overall occurrence and number of papers including the word at least once (B).

results (*Sandve et al., 2013*), others provide an all-embracing set of research instructions (*Stodden et al., 2016*; *Nosek et al., 2015*; *Gil et al., 2016*).

*Data.* A recurring aspect we encountered is making data accessible for other researchers (cf. *Reichman, Jones & Schildhauer, 2011*), ideally as archived assets having a Digital Object Identifier (DOI) and supplemented by structured metadata (*Gewin, 2016*). *Stodden et al. (2016)* consider legal aspects, such as sharing data publicly under an open license to clarify reusability. Further recommendations refer to modifying scientific practises, such as citation standards to ensure proper acknowledgement (*Nosek et al., 2015*), fostering data transparency (*McNutt, 2014*), and using open data formats to mitigate potentially disappearing proprietary software (*Gewin, 2016*). According to *Reichman, Jones & Schildhauer (2011)*, journals and funders should include data sharing in their guidelines.

*Methods.* A key requirement (*Sandve et al., 2013*) concerning methods is sharing used or developed software, where software should be published using persistent links (*Stodden et al., 2016*; *Gil et al., 2016*) and descriptive metadata (*Reichman, Jones & Schildhauer, 2011*). Similar to data, important concerns for software are open licensing (*Barba, 2016*) and proper credits (*Stodden et al., 2016*). Researchers can accomplish software transparency by using version control systems (cf. *Sandve et al., 2013*), and transparency mandates

**Table 1  Reproducibility-related keywords in the corpus, ordered by sum of matches per paper.** For full references of the corpus papers see Supplemental Material.

| Citation | Reproduc. | Replic. | Repeatab. | Code | Software | Algorithm(s) | (pre)process. | Data | Result(s) | All |
|---|---|---|---|---|---|---|---|---|---|---|
| Foerster et al. (2012) | 0 | 0 | 0 | 2 | 3 | 11 | 140 | 129 | 41 | 326 |
| Wiemann & Bernard (2014) | 0 | 0 | 0 | 0 | 0 | 0 | 20 | 98 | 3 | 123 |
| Mazimpaka & Timpf (2015) | 0 | 0 | 0 | 3 | 0 | 4 | 4 | 97 | 10 | 118 |
| Steuer et al. (2015) | 0 | 0 | 0 | 0 | 0 | 25 | 12 | 64 | 17 | 118 |
| Schäffer et al. (2010) | 0 | 0 | 0 | 0 | 10 | 1 | 26 | 65 | 6 | 108 |
| Rosser et al. (2016) | 0 | 0 | 0 | 0 | 2 | 1 | 42 | 51 | 6 | 105 |
| Gröchening et al. (2014) | 0 | 0 | 0 | 0 | 0 | 3 | 2 | 69 | 27 | 101 |
| Almer et al. (2016) | 0 | 0 | 0 | 1 | 1 | 1 | 22 | 53 | 22 | 100 |
| Magalhães et al. (2012) | 0 | 0 | 0 | 2 | 1 | 20 | 52 | 9 | 1 | 85 |
| Juhász & Hochmair (2016) | 0 | 0 | 0 | 0 | 1 | 1 | 2 | 55 | 11 | 70 |
| Wiemann (2016) | 0 | 0 | 0 | 0 | 3 | 0 | 8 | 55 | 1 | 69 |
| Fan et al. (2014) | 0 | 0 | 0 | 0 | 0 | 3 | 8 | 44 | 12 | 67 |
| Merki & Laube (2012) | 0 | 0 | 0 | 0 | 0 | 9 | 6 | 40 | 6 | 62 |
| Zhu et al. (2017) | 2 | 2 | 0 | 2 | 0 | 10 | 7 | 32 | 6 | 61 |
| Kuhn & Ballatore (2015) | 0 | 0 | 1 | 2 | 14 | 1 | 5 | 26 | 8 | 58 |
| Soleymani et al. (2014) | 1 | 0 | 0 | 0 | 0 | 0 | 4 | 39 | 9 | 56 |
| Fogliaroni & Hobel (2015) | 0 | 0 | 0 | 0 | 0 | 3 | 14 | 30 | 5 | 52 |
| Osaragi & Hoshino (2012) | 0 | 0 | 0 | 0 | 0 | 0 | 5 | 36 | 7 | 48 |
| Stein & Schlieder (2013) | 0 | 0 | 0 | 0 | 0 | 0 | 3 | 42 | 3 | 48 |
| Körner et al. (2010) | 0 | 0 | 0 | 0 | 0 | 6 | 5 | 30 | 4 | 45 |
| Knoth et al. (2017) | 0 | 0 | 0 | 3 | 2 | 1 | 6 | 25 | 7 | 44 |
| Raubal & Winter (2010) | 0 | 0 | 0 | 1 | 1 | 1 | 18 | 0 | 13 | 34 |
| Konkol et al. (2017) | 1 | 0 | 0 | 3 | 1 | 1 | 2 | 4 | 19 | 31 |
| Kiefer et al. (2012) | 1 | 0 | 0 | 0 | 2 | 1 | 9 | 10 | 8 | 31 |
| Haumann et al. (2017) | 0 | 0 | 0 | 0 | 0 | 6 | 8 | 10 | 2 | 26 |
| Josselin et al. (2016) | 0 | 0 | 0 | 0 | 2 | 1 | 9 | 5 | 8 | 25 |
| Heinz & Schlieder (2015) | 1 | 0 | 0 | 2 | 1 | 3 | 2 | 14 | 2 | 25 |
| Osaragi & Tsuda (2013) | 0 | 0 | 0 | 1 | 1 | 0 | 3 | 16 | 2 | 23 |
| Baglatzi & Kuhn (2013) | 1 | 0 | 0 | 0 | 0 | 0 | 6 | 12 | 3 | 22 |
| Scheider et al. (2014) | 0 | 0 | 0 | 0 | 1 | 0 | 0 | 13 | 4 | 19 |
| Brinkhoff (2017) | 0 | 0 | 0 | 0 | 1 | 9 | 2 | 3 | 2 | 17 |
| Schwering et al. (2013) | 0 | 0 | 0 | 0 | 0 | 4 | 2 | 3 | 5 | 14 |
| **Total** | 7 | 2 | 1 | 22 | 47 | 126 | 454 | 1,179 | 280 | 2,131 |

using open source instead of proprietary software (*Steiniger & Hay, 2009*). Since full computational reproducibility can depend on exact software versions (*Gronenschild et al., 2012*), the computational environment needs to be reported (cf. *Stodden et al., 2016*; *Gil et al., 2016*). Further software-specific recommendations are workflow tracking (*Stodden & Miguez, 2014*) and keeping a record of analysis parameters (*Gil et al., 2016*). *Sandve et al. (2013)* suggest avoiding manual data manipulation steps and instead using scripts to increase transparency in data preprocessing.

*Results.* *Sandve et al. (2013)* focus on results-related guidelines such as storing intermediate results and noting seeds if computations include randomness. Journals should conduct a reproducibility check prior to publication (*Stodden et al., 2016*) or funding should be explicitly granted for making research results repeatable (*Collberg & Proebsting, 2016*). Finally, *Barba (2016)* describes the contents and benefits of a "reproducibility package" to preserve results.

*Structure.* While the papers discussed above focus on specific aspects of reproducibility, an overarching structure for all facets of research can provide important context. But none of the suggestions for packaging workflows are widely established, for example *Gentleman & Lang (2007)* use programming language packaging mechanisms, *Bechhofer et al. (2013)* Linked Data, or *Nüst et al. (2017)* nested containers.

*Section summary.* Most recommendations and suggestions to foster open and reproducible research address data and methods. Particularly, methods cover a broad range of aspects including recommendations on data preprocessing, the actual analysis, and the computational environment. Results receive less attention, possibly because they are strongly connected with other aspects. While most of the recommendations address authors, only few target journals and research institutions.

### Definition and criteria

This paper focuses on reproducibility in the context of conference publications and adopts the described consensus (see 'Related work') for the following definition.

> A reproducible paper ensures a reviewer or reader can recreate the computational workflow of a study or experiment, including the prerequisite knowledge and the computational environment. The former implies the scientific argument to be understandable and sound. The latter requires a detailed description of used software and data, and both being openly available.

We build on the recommendations from 'Recommendations and suggestions in literature' and differentiate data, methods, and results as separate dimensions of reproducibility. We conceptualised each reproducibility dimension as a criterion, and for each criterion, we developed levels of attained reproducibility. In order to increase reproducibility of this study and improve inter-rater agreement, we created a corresponding rubric that explains the requirements. Together, the three criteria and their levels address specifics of GIScience research and allow for a fine-grained assessment of reproducibility.

However, early during the evaluation process, it became clear that the assessed corpus papers showed great variation in data, methods, and type of results. For example, data used during the reported studies varies from spatial data to qualitative results from surveys. Methods are particularly diverse, ranging from the application of spatial analysis operations to statistical approaches or simulations. Results include maps, formulas, models or diagrams. Therefore, we decided to split the methods criterion into three sub-criteria addressing the distinct phases and respective software tools: data preprocessing, analysis

methods and workflows, and computational environment. Following this change, we re-evaluated already assessed corpus papers.

Figure 2 shows the reproducibility criteria for each of the categories *Data*, *Methods*, and *Results*, and their levels. The levels are either *not applicable* (NA) or range from *no* (value of 0) to *full* (3) reproducibility. The level 0 unavailable means that there is insufficient documentation in the paper, or the information is only available upon request (since this cannot be guaranteed and we could not check availability for all studies). The level 1 documented means that there is sufficient information to recreate at least parts of the study, but no concrete data or code or models. Level 2 available means that required data and code is provided. Finally, level 3 available, open and permanent, adds the requirement of unrestricted and sustainable access, e.g., through permanent links to open repositories containing data, relevant methods and workflows (such as software versions, hardware specifications, scripts), and all results (including intermediary ones or those not discussed in detail in the study). The *Methods* criteria do not include the "permanent" aspect, because there is no suitable single identifier to define the complex properties of code, libraries and system environment, although a DOI may be used to collectively identify all these items as source or binary files. Licensing is important for reproducibility, because only a clear license, which ideally is well-known and established, allows use of copyrighted content. So in this sense "open" means "open enough for reproduction", but in practice the used licenses should be fully open and allow modification and sharing beyond mere access and use[3].

The intermediate levels (1 and 2) allow a differentiated evaluation. For example for data at level 1, data is not accessible but documented sufficiently, so others can recreate it; at level 2 data is available yet in a non-persistent way or with a restrictive license. The requirements are cumulative, meaning that higher levels of reproducibility include lower levels' requirements. The reproducibility rubric was developed in iterative discussions between all raters, using the examined literature on reproducibility as point of reference.

By design, our criteria cannot be applied to conceptual research publications, namely those without data or code. Their evaluation is covered by an editorial peer review process (see for example *Ferreira et al. (2016)* for history and future of peer review), and assessing the merit of an argument is beyond the scope of this work.

## Author feedback on assessment of reproducibility (survey)

To better understand the reasons behind the scores and to give the authors an opportunity to respond after the reproducibility of their research was assessed, we designed a survey using Google Forms (https://www.google.com/forms/about/) (see Table 2, cf. *Baker (2016a)* for a large scale survey on the topic). The full survey, as it was shown to the participants, is included in the Supplemental Material.

Along with the survey, authors were provided with the results of our evaluation of their specific papers, and they were asked to express their agreement or disagreement with the results. The four main questions of the survey were designed to find out whether authors considered reproducibility important in the first place, and if so, what prevented them

[3]Cf. The Open Definition, https://opendefinition.org/: "Open data and content can be freely used, modified, and shared by anyone for any purpose".

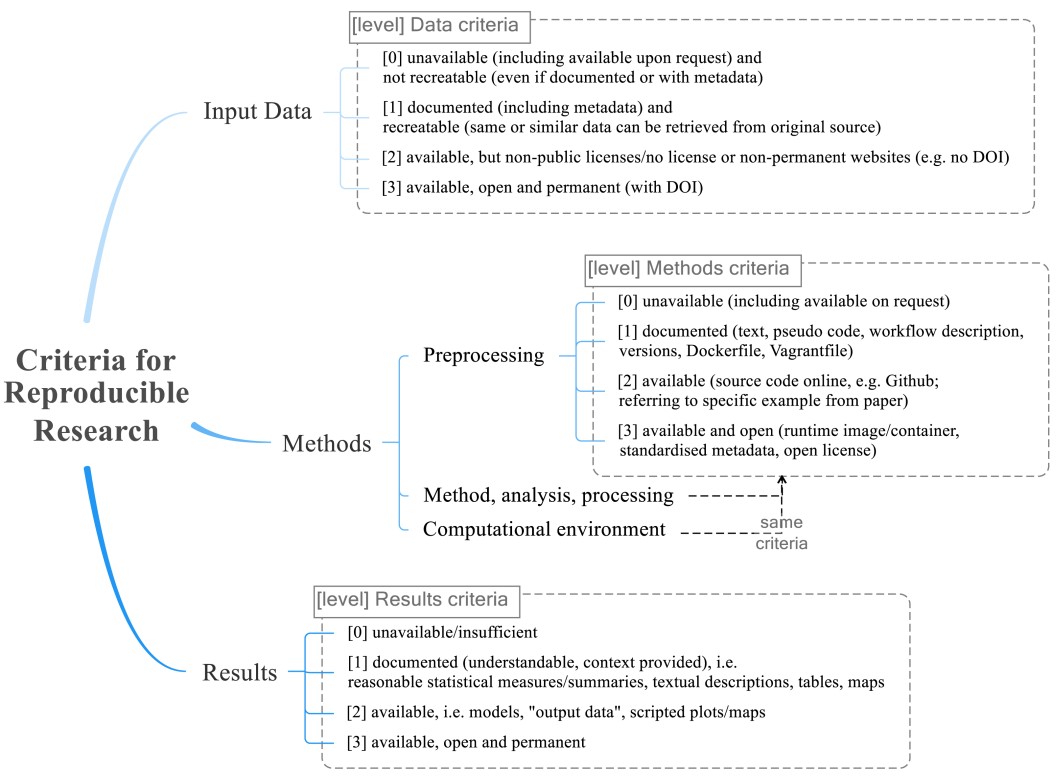

**Figure 2** **The final reproducible research criteria used for the evaluation.** The categories *Data*, *Methods* (sub-categories: preprocessing, method/analysis/processing, and computational environment), and *Results* each have four levels ranging from 0 = not reproducible to 3 = fully reproducible.

from fully achieving it. Finally, the authors were asked to provide their own opinion and suggestions for the AGILE community to encourage publishing fully reproducible papers.

## RESULTS

### Assessment of reproducibility

To address RQ 3, we reviewed the papers in the corpus with the introduced criteria. Our objective in publishing the full evaluation results is not to criticise or rank individual papers, but to identify the current overall state of reproducibility in GIScience research in a reproducible manner. The scientific merit of all papers was already proven by their nomination for the best paper award.

The procedure was as follows: First, we determined a maximum number of papers for a single evaluator to reach two evaluators per paper. Second, we grouped evaluators according to their affiliation or research group. Evaluators then chose to review papers without a conflict of interest on a first come first served basis until two goals were achieved: the evaluator reached her maximum number of reviews and two evaluators from different research groups reviewed the paper. For assigning a level of reproducibility, the general guideline was to apply the lower of two possible levels in cases of doubt, such as partial

**Table 2** Survey questions (except for paper identification questions; for full questionnaire see Supplemental Material).

| Question | Possible answers |
|---|---|
| 1. Have you considered the reproducibility of research published in your nominated paper? | • Yes, it is important to me that my research is fully reproducible<br>• Yes, I have somewhat considered reproducibility<br>• No, I was not concerned with it<br>• Other (please add) |
| 2. Do you agree with our rating of your publication? Please comment. | *Open answer* |
| 3a. Please rate how strongly the following circumstances have hindered you from providing all data, methods and results used/developed during your research? | • The need to invest more time into the publication<br>• Lack of knowledge how to include data/methods/results into the publication<br>• Lack of tools that would help to attach data/methods/results to the publication<br>• Lack of motivation or incentive<br>• Legal restrictions<br>Available ratings:<br>• Not at all<br>• Slightly hindered<br>• Moderately hindered<br>• Strongly hindered<br>• Main reason |
| 3b. Please add here if there were any other hindering circumstances | *Open answer* |
| 4. What would you suggest to AGILE community to encourage publishing fully reproducible papers? | *Open answer* |

fulfilment of a criterion or disagreement between the evaluators. All reviewers discussed disagreements and open questions after an initial round of evaluation comprising one to three reviews per researcher, and after completing all reviews. Because the assessment focuses on algorithmic and data-driven research papers, five fully conceptual papers were not assessed, while 15 partly conceptual ones were included. Notably, the data preprocessing criterion did not apply to 14 research papers. Table 3 shows the assessment's results.

Figure 3 shows the distribution of reproducibility levels for each criterion. None of the papers reach the highest level of reproducibility in any category. Only five papers reach level 2 in the data criterion, which is still the highest number of that level across all categories. Especially problematic is the high number of papers (19) with level 0 for data, meaning that the specific data is not only unavailable but it is not re-createable from the information in the paper. Data preprocessing applies to 18 publications, and the levels are low. Only one publication has level 2. Concerning the methods and results criteria, 19 out of 32 papers have level 1 in both, meaning an understandable documentation is provided in the text.

Figure 4 shows that average reproducibility levels are low and do not change significantly over time, with the mean over all categories being below level 1 for all years. The categories are ordinal variables, meaning they have an implicit order but an unknown "distance" between them. They can be compared (3 is higher than 2), but absolute differences in numbers must not be interpreted. Moving one level up from 0 to 1 is not the same as from 2 to 3. Averaging on ordinal variables must be conducted with care: Mode and median

**Table 3  Reproducibility levels for paper corpus; '-' is category not available.** For full references of the corpus papers see Supplemental Material.

| Author | Short paper | Input data | Preprocessing | Method/analysis/ processing | Computational environment | Results |
|---|---|---|---|---|---|---|
| Zhu et al. (2017) | | 0 | 1 | 1 | 1 | 1 |
| Knoth et al. (2017) | | 0 | – | 0 | 1 | 1 |
| Konkol et al. (2017) | | 2 | 2 | 1 | 1 | 1 |
| Haumann et al. (2017) | X | 0 | 1 | 1 | 0 | 1 |
| Brinkhoff (2017) | X | 0 | – | 1 | 0 | 0 |
| Almer et al. (2016) | | 0 | – | 1 | 1 | 1 |
| Wiemann (2016) | | 2 | – | 1 | 1 | 1 |
| Juhász & Hochmair (2016) | | 0 | 1 | 1 | 0 | 0 |
| Josselin et al. (2016) | X | 1 | – | 0 | 0 | 1 |
| Rosser et al. (2016) | X | 0 | – | 1 | 0 | 0 |
| Kuhn & Ballatore (2015) | | – | – | – | – | – |
| Mazimpaka & Timpf (2015) | | 2 | 1 | 1 | 1 | 1 |
| Steuer et al. (2015) | | 2 | 0 | 1 | 1 | 1 |
| Fogliaroni & Hobel (2015) | X | – | – | – | – | – |
| Heinz & Schlieder (2015) | X | 0 | 0 | 1 | 1 | 1 |
| Scheider et al. (2014) | | 1 | 1 | 2 | 1 | 1 |
| Gröchening et al. (2014) | | 2 | 0 | 1 | 0 | 1 |
| Fan et al. (2014) | | 0 | 1 | 1 | 0 | 1 |
| Soleymani et al. (2014) | X | 0 | 0 | 1 | 0 | 0 |
| Wiemann & Bernard (2014) | X | 0 | 0 | 1 | 0 | 0 |
| Osaragi & Tsuda (2013) | | 0 | 1 | 1 | 0 | 1 |
| Baglatzi & Kuhn (2013) | | – | – | – | – | – |
| Li et al. (2013) | X | 0 | 0 | 1 | – | 1 |
| Stein & Schlieder (2013) | X | 0 | – | 1 | 0 | 1 |
| Osaragi & Hoshino (2012) | | 0 | 0 | 1 | 0 | 1 |
| Magalhães et al. (2012) | | 0 | 0 | 1 | 0 | 0 |
| Foerster et al. (2012) | | 1 | – | 1 | 1 | 1 |
| Merki & Laube (2012) | X | 0 | – | 1 | 1 | 1 |
| Kiefer et al. (2012) | X | 0 | 1 | 1 | 0 | 1 |
| Raubal & Winter (2010) | | – | – | – | – | – |
| Schäffer et al. (2010) | | 0 | 0 | 1 | 1 | 1 |
| Körner et al. (2010) | | – | – | – | – | – |

are mostly seen as acceptable averaging functions for ordinal data, while the mean is seen inapplicable by some.

We decided not to use median or mode, because they hide all differences between the categories. The mean should not be applied for a single paper, whereby all categories in a single paper are averaged, because different evaluation rules would be combined into a meaningless number. Being aware of these limitations and the small dataset size, we opted to apply the mean and a statistical summary to categories to compare values between the different categories, and to compare the two large groups within the paper corpus (full and short papers).

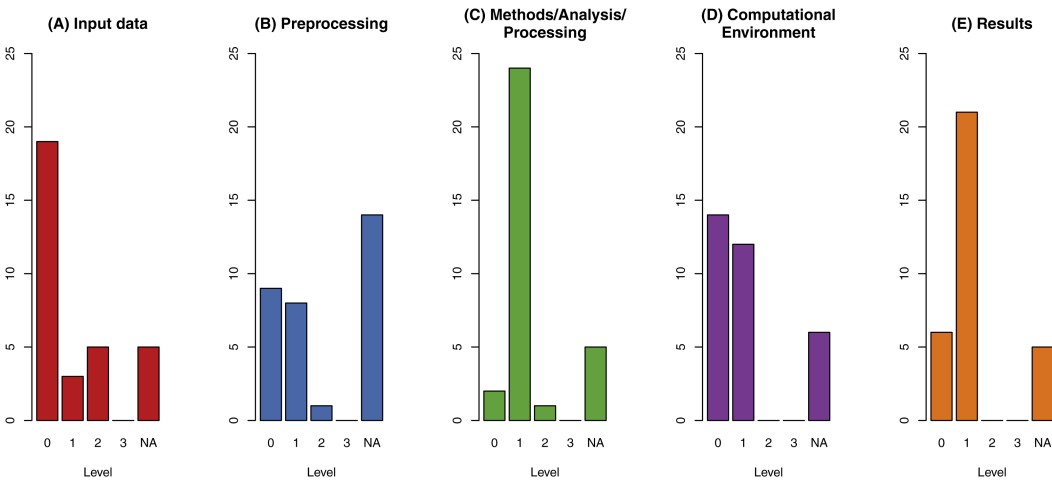

**Figure 3    Results of reproducibility assessment across all categories for the assessment of reproducibility:** *Data* **(A),** *Methods* **with sub-categories preprocessing (B), method/analysis/processing (C) and computational environment (D), and** *Results* **(E).**  The level of reproducibility ranges from 0 (not reproducible) to 3 (fully reproducible); NAs include 5 conceptual papers (all categories are NA).

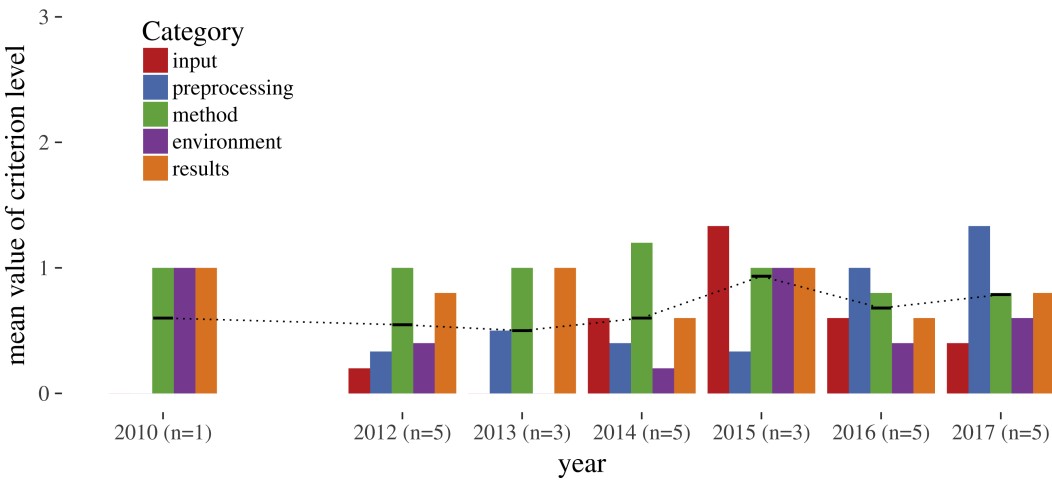

**Figure 4    Mean reproducibility levels per category over time; black dotted line connects the mean per year over all categories (in 2010 only one of three papers could be assessed, reaching level 1 for methods).**

Tables 4 and 5 contain summary statistics per criterion and means for full and short papers. For each criterion, full papers reach higher levels of reproducibility than short papers (see Table 5).

## Author feedback (survey)

The full survey responses are included in this paper's repository (*Nüst, 2018*). The survey was sent to authors via e-mail and was open from 23 October to 24 November 2017. In case of obsolete e-mail addresses, we searched for updated ones and resent the form. Out

**Table 4** Statistics of reproducibility levels per criterion.

|        | Input data | Preproc. | Method/analysis/proc. | Comp. env. | Results |
|--------|-----------|----------|----------------------|-----------|---------|
| Min.   | 0.00 | 0.00 | 0.00 | 0.00 | 0.00 |
| Median | 0.00 | 0.50 | 1.00 | 0.00 | 1.00 |
| Mean   | 0.48 | 0.56 | 0.96 | 0.46 | 0.78 |
| Max.   | 2.00 | 2.00 | 2.00 | 1.00 | 1.00 |
| NA's   | 5.00 | 14.00 | 5.00 | 6.00 | 5.00 |

**Table 5** Mean levels per criterion for full and short papers.

|              | Input data | preproc. | Method/analysis/proc. | Comp. env. | Results |
|--------------|-----------|----------|----------------------|-----------|---------|
| Full papers  | 0.75 | 0.67 | 1.00 | 0.62 | 0.88 |
| Short papers | 0.09 | 0.33 | 0.91 | 0.20 | 0.64 |

of a total of 82 authors, 22 filled in the survey, resulting in responses for 17 papers, because six participants did not give consent to use their answers, two authors participated twice for different papers, and some papers had more than one individual response.

Authors were asked to comment on whether they agreed or disagreed with our evaluations of their specific paper. Seven responses fully agreed with the evaluation, five agreed partly, two expressed disagreement, and one did not answer the question. Most disagreements addressed the definition of criteria. Multiple authors argued that such requirements should not be applicable for short papers, and that specific data is not always necessary for reproducibility. Others disagreed about treating "availability upon request" as "unavailable". One argued that OpenStreetMap data is by default "open and permanent", but for our criteria citing OpenStreetMap lacked direct links to specific versioned subsets of data.

The answers suggest that authors are generally aware of the need for reproducibility and in principle know how to improve it in their work. However, many do not consider it a priority, saying that they did not incorporate reproducibility because of a lack of motivation (eight respondents) or the required extra effort required, which they say is disproportionately large in comparison to the added value.

According to the survey results, reproducibility was important to more than half of the respondents (see Fig. 5). Only two respondents claimed they were not at all concerned about it (both short papers). Further comments revealed that some authors consider short papers as introductions of new concepts and generally too short for reproducibility concerns. The paper corpus supports this opinion because short papers reach overall lower reproducibility levels.

In contrast, we argue that transparency should not depend on the publication type but is a feature of the entire scientific process. Especially at early stages, the potential for external review and collaboration can be beneficial for authors. Further, putting supplementary materials in online repositories addresses the problem of word count limits (for full and short papers), which many authors struggle with.

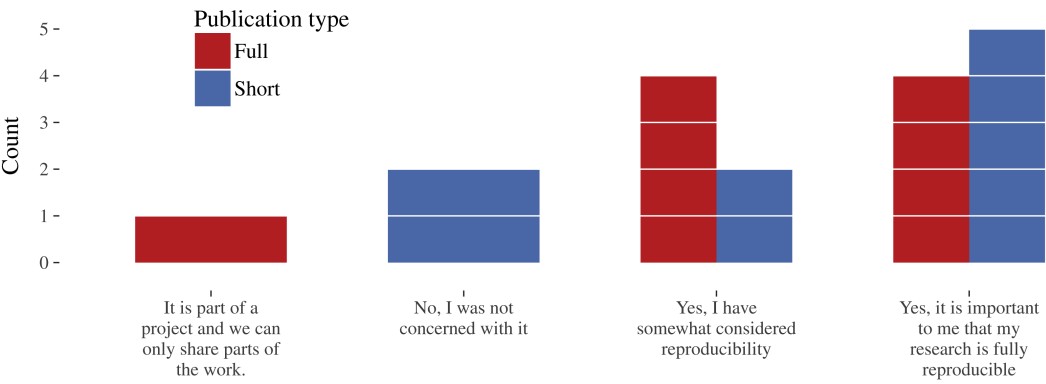

Have you considered the reproducibility of research published in your nominated paper? (n = 18)

**Figure 5** **Author survey results on the importance of reproducibility.**

To identify barriers to reproducibility, the authors were asked to rate how strongly five predefined barriers (Table 2) impacted their work's reproducibility. They could also add their own reasons, for which they mentioned paper length restrictions and the need for additional financial resources. Table 6 shows that the most frequently mentioned reasons were legal restrictions and lack of time, where only one respondent indicated that these factors played no role. Although lack of knowledge on how to include data, methods and results was not considered by many as a barrier, several respondents noted a lack of supporting tools as a main impediment for reproducibility.

Respondents also shared their ideas for how AGILE could encourage reproducibility in its publications. Four suggested Open Access publishing and asked for solutions to deal with sensitive data. A few suggested encouraging and promoting collaboration across research institutes and countries to mitigate ephemeral storage and organisations. Some respondents proposed that an award could be given for reproducible papers, reproducibility could be required for the best paper nomination, or conference fees could be waived for reproducible papers. In summary, almost all authors agreed on the importance of the topic and its relevance for AGILE.

# DISCUSSION

## A critical review of this paper's reproducibility

We acknowledge this paper has its own shortcomings with respect to reproducibility. The data, code, and a description of the runtime environment are transparently published on GitHub (https://github.com/nuest/reproducible-research-and-giscience) and deposited in an open repository under the DOI https://doi.org/10.5281/zenodo.1227260 (*Nüst, 2018*). The repository comprises an anonymised table with the survey results and a literate programming document, which transparently combines data preprocessing, analysis, and visualisations. The runtime environment description is based on Docker (https://en.wikipedia.org/wiki/Docker_(software)) and allows readers to easily open an

**Table 6  Hindering circumstances for reproducibility for each survey response ($n = 17$) sorted by barrier type for the category with most "Main reason" occurences; each line is one response and background colour corresponds to cell text.**

| Legal restrictions | Lack of time | Lack of tools | Lack of knowledge | Lack of incentive |
| --- | --- | --- | --- | --- |
| Main reason | Strongly hindered | Not at all | Not at all | Strongly hindered |
| Main reason | Not at all | Not at all | Not at all | Moderately hindered |
| Main reason | Slightly hindered | Strongly hindered | Moderately hindered | Strongly hindered |
| Main reason | Not at all | Slightly hindered | Not at all | Not at all |
| Strongly hindered | Strongly hindered | Strongly hindered | Moderately hindered | Strongly hindered |
| Moderately hindered | Main reason | Not at all | Not at all | Not at all |
| Slightly hindered | Moderately hindered | Slightly hindered | Slightly hindered | Moderately hindered |
| Slightly hindered | Not at all | Main reason | Strongly hindered | Not at all |
| Not at all | Moderately hindered | Not at all | Moderately hindered | Not at all |
| Not at all | Strongly hindered | Strongly hindered | Strongly hindered | Slightly hindered |
| Not at all | Moderately hindered | Not at all | Not at all | Not at all |
| Not at all | Slightly hindered | Main reason | Not at all | Strongly hindered |
| Not at all | Main reason | Not at all | Not at all | Not at all |
| Not at all | Main reason | Not at all | Not at all | Not at all |
| Not at all | Moderately hindered | Moderately hindered | Not at all | Strongly hindered |
| Not at all | Not at all | Not at all | Not at all | Not at all |
| Not at all | Slightly hindered | Not at all | Slightly hindered | Not at all |

interactive analysis environment in their browser based on Binder (http://mybinder.org/, (cf. *Holdgraf, 2017*). The working link to launch the binder is https://mybinder.org/ v2/gh/ nuest/reproducible-research-and-giscience/6 and the file README.md provides instructions on the usage. The *input data* (i.e., the paper corpus) for the text analysis cannot be re-published due to copyright restrictions. Our sample is biased (although probably positively), as we only considered award nominees. Access to all papers would have allowed a random sample from the population. Regarding the *method*, the created criteria and how they were assigned by humans cannot honour all details and variety of individual research contributions and is inherently subjective. We tried to mitigate this by applying a "four eyes" principle, and transparently sharing internal comments and discussion on the matter in the code repository. Using our own classification, we critically assign ourselves level 0 for data and level 3 for methods and results.

## Improving day-to-day research in GIScience

Our evaluation clearly identifies issues of reproducibility in GIScience. Many of the evaluated papers use data and computer-based analysis. All papers were nominated for the best paper award within a double-blind peer review and thus represent the upper end of the quality spectrum at an established conference. Yet, overall reproducibility is low and no positive trend is perceivable. It seems that current practises in scientific publications lack full access to data and code. Instead, only methods and results are documented in writing.

In order to significantly improve the reproducibility of research, there must be changes in educational curricula, lab processes, universities, journal publishing, and funding agencies

(*Reproducible Research, 2010*; *McKiernan, 2017*) as well as reward mechanisms that go beyond paper citations (cf. term "altmetrics" in *Priem et al., 2010*). This is a major and long-term endeavour. Here, we focus on recommendations and suggestions for individual researchers and a specific organisation: AGILE. A snowball effect may lead to a change in practises in the GIScience community. The remainder of this paper addresses RQ 4 by formulating suggestions to researchers and the AGILE conference organisers.

## Suggestions to authors

Regarding habits and workflows, the Carpentries (the union (http://www.datacarpentry.org/blog/merger/) of Data Carpentry (*Teal et al., 2015*) and Software Carpentry (*Wilson, 2006*)) offer lessons on tools to support research, such as programming and data management, across disciplines. Further resources are available from programming language and software communities, research domains, and online universities. Often these resources are available for free because the software is Free and Open Source Software (FOSS) and driven by a mixed community of users and developers. Ultimately, proprietary software is a deal-breaker for reproducibility (cf. *Ince, Hatton & Graham-Cumming, 2012*; *Baker, 2016b*). OSGeo-Live (https://live.osgeo.org/) provides a simple environment to test open alternatives from the geospatial domain, and several websites offer help in finding FOSS comparable to commercial products (e.g., https://opensource.com/alternatives or https://alternativeto.net). But, authors can do more than just use open software to improve reproducibility. It is not only about the software. They can engage in simple tasks such as "naming things" sensibly (https://speakerdeck.com/jennybc/how-to-name-files by Jennifer Bryan), they can be realistic by not striving for perfection but following "good enough practices in scientific computing" (*Wilson et al., 2017*), they can explore "selfish reasons to work reproducibly" (*Markowetz, 2015*), and they can follow FAIR[4] guidelines with "structuring supplemental material" (*Greenbaum et al., 2017*).

## Recommendations to conferences in GIScience and organisations like AGILE

*What can conferences and scientific associations do to encourage reproducibility?* A crucial step in improving reproducibility of GIScience research is acknowledging the important role organisations like AGILE can play in adopting reproducible research practises, which can be built upon a large body of guidelines, documentation and software. In the remainder of this section we propose concrete actions for organisations, using AGILE as the leading example.

AGILE could show that it recognizes and supports reproducibility by offering an **award for reproducible papers**. This is already done by other communities, e.g., the ACM SIGMOD 2017 Most Reproducible Paper Award (http://db-reproducibility.seas.harvard.edu/ and https://sigmod.org/2017-reproducibility-award/). At AGILE, when reviewers suggest submissions to be nominated for best (short) papers, , they could also have these papers briefly checked for reproducibility. This check could be performed by a new Scientific Reproducibility Committee led by a Reproducibility Chair, working alongside the existing committees and their chairs. Committee membership would be publicly recognised. The

[4]Force11.org. Guiding principles for findable, accessible, interoperable and re-usable data publishing: version B1.0. https://www.force11.org/node/6062

"most reproducible paper" could be prominently presented in the conference's closing session.

*Kidwell et al. (2016)* demonstrate that open data **badges** have had a positive effect on actual publishing of data in the journal *Psychological Science*, which uses badges and corresponding criteria from the Center for Open Science (https://osf.io/tvyxz/wiki/home/) (COS). Further examples are the "kite marks" used by the journal *Biostatistics* (*Peng, 2011*), the common standards and terms for artifacts used by the Association for Computing Machinery's (ACM) (https://www.acm.org/publications/policies/artifact-review-badging), and the Graphics Replicability Stamp Initiative (GRSI) (http://www.replicabilitystamp.org/). While AGILE could invent its own badges, re-using existing approaches has practical advantages (no need to design new badges), organisational advantages (no need to reinvent criteria), and marketing advantages (higher memorability). Author guidelines would include instructions on how to receive badges for a submission. The evaluation of badge criteria would be integrated in the review and could inform the reproducible paper award.

**Author guidelines** are the essential place to set the foundation for a reproducible conference (cf. SIGMOD 2018 CFP, https://sigmod2018.org/calls_papers_sigmod_research.shtml). Independently of advertising awards and badges, author guidelines should include clear guidelines on when, how, and where to publish supplemental material (data, code). *Author guidelines for computational research* must make authors aware to highlight reproducibility-related information for reviewers and readers. These guidelines should contain practical advice, such as code and data licenses[5], and instructions on how to work reproducibly, such as by providing a space for sharing tools and data, which is the most popular suggestion from the survey (seven respondents).

While the established peer-review process works well for conceptual papers, a special **track or submission type**[6] could accommodate submissions focussing on reproducibility without an original scientific contribution and an adapted process (e.g., public peer review). Such publications can include different authors, e.g., technical staff, or even reviewers as practised by Elsevier's *Information Systems* journal. Publications in a special track can also mitigate limitations on research paper lengths. Unfortunately, they can also convey the counterproductive message of reproducibility being cumbersome and uncommon.

Submissions through this special track as well as the regular conference proceedings should be published as **Open Access** (see https://open-access.net/DE-EN/information-on-open-access/open-access-strategies/) content in the future. It might even be possible to re-publish short papers and abstracts of previous conferences after solving juridical concerns (e.g., if author consent is required). To do this, AGILE could utilise existing repositories or operate its own, where using third party repositories[7] for supplements would reduce the burden on the AGILE organisation. Choosing **one repository** allows for collecting all AGILE submissions under one tag or community (cf. http://help.osf.io/m/sharing/l/524053-tags and https://zenodo.org/communities/). An AGILE-specific repository would allow more control, but would require more work and might have lower visibility, since the large repositories are well indexed by search engines. Both approaches would support a double-blind review by providing anonymous view-only copies of supplemental material (see http://help.osf.io/m/links_forks/l/524049-create-a-view-only-link-for-a-project).

---

[5]E.g., OSI compliant for code and Open Definition compliant for data, see http://licenses.opendefinition.org/.

[6]See IEEE's CiSE magazine's Reproducible Research Track https://www.computer.org/cise/2017/07/26/reproducible-research-track-call-for-papers/, and Elsevier journal Information Systems' section for invited reproducibility papers, https://www.elsevier.com/journals/information-systems/0306-4379/guide-for-authors.

[7]Beside the incumbents Figshare (https://figshare.com/), Open Science Framework (OSF) (https://osf.io/, community-driven) and Zenodo (https://zenodo.org/, potentially preferable given AGILE's origin because it is funded by EU), a large number of Open Access repositories exists, see http://roar.eprints.org/ and http://opendoar.org/, including platforms by publishers, e.g., Springer (https://www.springer.com/gp/open-access), or independent organisations, e.g., LIPIcs proceedings (https://www.dagstuhl.de/en/publications/lipics)

We see AGILE, carried by its member labs and mission (https://agile-online.org/index.php/about-agile), as being in a unique position among GIScience conferences to establish a common understanding and practise of reproducible research. Firstly, member labs can influence education, especially at the graduate level, and ideally collaborate on **open educational material**. Completing a Ph.D. in an AGILE member lab and participating in AGILE conferences should qualify early career scientists to publish and review reproducible scholarly works. Secondly, the conference can take a leading role in setting up new norms for conference review and publication but at the same time cooperate with other conferences (e.g., ACM SIGMOD). At first AGILE would encourage but eventually demand the highest level of reproducibility for all submissions. This process certainly will take several years to complete.

## CONCLUSIONS

*What skills related to reproducibility are desirable for authors at GIScience conferences in 2028?* Predicting 10 years ahead might not be scientific, but it allows for formulating a vision to conclude this work. We assume that in 10 years, hardly any paper will not utilise digital methods, such as software for analysis, interactive visualisations, or open data. Ever more academics will face competitive selection processes, where quality of research will be measured by its transparency and novelty. To achieve novelty in a setting where all research is saved, findable and potentially interpreted by artificial intelligence (*Jones, 2016*), a new contribution must be traceable. Thus, the trend towards Open Science will be reinforced until it is standard practise to use and publish open source code and open data as well as to incorporate alternative metrics beyond citations. As of now, AGILE is not ready for such research. It has identifiers (DOIs) only for full publications and lacks open licenses for posters and (short) papers. Statements on preprints (publication before submission) and postprints ("green" Open Access, see https://open-access.net/DE-EN/information-on-open-access/open-access-strategies/) are missing.

Researchers, conference organisers, and programme committees will have to leave their comfort zone and change the way they work. Also, in order to overcome old habits, they will have to immediately see the benefits of the new ways (*Wilson et al., 2017*). The evidence for benefits of Open Science are strong (*McKiernan et al., 2016*), but to succeed, the community must embrace the idea of a reproducible conference. We acknowledge that fully reproducible GIScience papers are no small step for both authors and reviewers, but making them the standard would certainly be a giant leap for GIScience conferences. We are convinced a conference like AGILE can provide the required critical mass and openness, and we hope the experiences and information provided in this work represent a sound starting point.

## ACKNOWLEDGEMENTS

We thank all authors who participated in the survey and the reviewers for their detailed comments and valuable suggestions for the manuscript. We would like to thank Celeste R.

Brennecka, from the Science Writing Support Service of the University of Münster, for her editorial support.

### Funding

Carlos Granell is funded by the Ramón y Cajal Programme of the Spanish government (grant number RYC-2014-16913). Daniel Nüst and Markus Konkol are supported by the project Opening Reproducible Research (https://www.uni-muenster.de/forschungaz/project/9520) funded by the German Research Foundation (DFG) under project numbers PE 1632/10-1 respectively KR 3930/3-1. The funders had no role in study design, data collection and analysis, decision to publish, or preparation of the manuscript.

### Grant Disclosures

The following grant information was disclosed by the authors:
Ramón y Cajal Programme of the Spanish government: RYC-2014-16913.
German Research Foundation (DFG): PE 1632/10-1, KR 3930/3-1.

### Competing Interests

Barbara Hofer is a member of the AGILE council (https://agile-online.org/index.php/community/council).

### Author Contributions

- Daniel Nüst, Barbara Hofer and Rusne Sileryte analyzed the data, contributed reagents/materials/analysis tools, prepared figures and/or tables, authored or reviewed drafts of the paper, approved the final draft.
- Carlos Granell analyzed the data, prepared figures and/or tables, authored or reviewed drafts of the paper, approved the final draft.
- Markus Konkol and Frank O. Ostermann analyzed the data, authored or reviewed drafts of the paper, approved the final draft.
- Valentina Cerutti analyzed the data, approved the final draft.

### Data Availability

GitHub: https://github.com/nuest/reproducible-research-and-giscience/
Zenodo: https://doi.org/10.5281/zenodo.1227260.

### Supplemental Information

Supplemental information for this article can be found online at http://dx.doi.org/10.7717/peerj.5072#supplemental-information.

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
