# Peer review of "Reproducible research and GIScience: an evaluation using AGILE conference papers"

_PeerJ, doi:10.7717/peerj.5072_

## Round 0.1 · original submission · Major Revisions

First I would like to apologize for the unusual long delay in the review but some external events prevent one of the reviewer to send a timely review.

Based on the two reviews, I think the paper is worth to be published provided you take into account the recommendation (or give arguments why you prefer not to take some into account if this is the case). I think we (editor + reviewers) are all supportive of your work and our goal is really to help you improve the manuscript. The major revision recommendation is merely an option for us to make sure reviewers can have an eye on your revised manuscript.

The "Binder" proposition might be worth to be investigated since it does not seem straightforward to run the provided code. In any case, an improved installation file would definitely help a reader to run the code.

Concerning the English level, and being a non-native English speaker myself, I would only recommend to have the revised manuscript to be read by a native speaker IF you have such possibility. Else, a thorough re-read by yourselves will do.

I hope you'll find the reviews useful. There isn't a hard deadline for sending the revision but the sooner the better (your manuscript will still be fresh in our memory).

·

Basic reporting

This article describes an empirical evaluation of the reproducibility of research in geographic information science (GIS), based on a sample of papers that received a "best paper award" at an AGILE conference and on a survey sent to the authors of these papers. The article is well written, using a clear language and overall good illustrations. The references to the literature on reproducibility show that the authors are well aware of the state of the art of research in this field.

In the supplemental material, the authors provide raw data for their study except where this is prevented by copyright restrictions. These restrictions are particularly important in this case because the main subject of the study is a set of published articles. The authors also provide an R implementation of their computational analyses. The raw data that is supplied is understandable in the context of the article, and could be used as the basis for further studies.

I have two requests for improvement on basic reporting:

1. The authors should add a short README file to the supplemental material, containing a reference to the paper and a short summary for each file, much like the description available on the PeerJ Web site.

2. The authors should add a legend to Table 6, explaining what the color codes mean. It looks like the colors merely reflect the text in the fields, but it is difficult to be certain about this.

Experimental design

The work seems well within the scope of PeerJ Computer Science, although I think the editors are more competent to make such a judgement than a humble reviewer. Four relevant research questions are clearly identified at the start, and answered one by one in the manuscript. The authors have applied the high level of rigor one would expect in the context of reproducible research to their own work. I hope that this solid study will encourage others to apply the same methodology to other areas of science.

In spite of the very good overall quality of this study, there are two points which in my opinion deserve more attention:

1. The authors introduce "reproducibility levels" represented by the integers 0 to 3, with the understanding that a higher number indicates more or better reproducibility. These levels are assigned to different aspects of each paper by a panel of human judges. Their definition (Figure 2) seems overall reasonable, but some questions remain. Why does licensing enter into the criteria for data but not for code? Why does licensing matter for reproducibility at all? Why does a permanent ID (DOI) matter for data and results, but not for code? Why does openness enter at all, given that reuse is a concern distinct from reproducibility? The answers to these questions don't make much of a difference to the results of this study, since the observed reproducibility levels are rather low, but since the authors' criteria are likely to be reused by others in the future, improved clarity is highly desirable.

2. Starting at line 232, the reproducibility levels are elevated to a metric for use in quantitative statistical analyses (Figure 4, Tables 4 and 5). This step seems problematic and should be justified. For example, what is the interpretation of an average reproducibility level of 0.7? What would an increase of 0.3 in a reproducibility level mean in practice? In what sense is an increase from 0.7 to 0.3 "the same" as an increase from 2.7 to 3?

Validity of the findings

The conclusions provide answers to the initial research questions that are overall convincing. The analysis looks sound, the major weakness of a rather small sample size is ackowledged and discussed by the authors.

I have a single disagreement with the authors' conclusions. Starting from line 271, the authors self-evaluate the reproduciblity of their work and reach level 3 for methods and results. In case of acceptance of this paper, I would agree for the results. However, method reproducibility at level 3 would require the availability of the computational environment, e.g. in the form of a Docker image.

This is not just a theoretical concern because I have not been able to run the R code supplied with this study, meaning that I cannot reproduce its results. It uses the library "wordcloud" whose dependency "slam" cannot be automatically installed from CRAN. Manual installation using instructions found on the Web (https://stackoverflow.com/questions/39885408/dependency-slam-is-not-available-when-installing-tm-package) failed on my machine (macOS 10.12) because it seems to require the run-time library of the gfortran compiler.

I have also attempted to read the R source code in the context of the provided explanations, in order to check if it corresponds to the method description in the manuscript. Unfortunately I have failed in this step as well. My knowledge of the R language is limited and superficial. In particular, I am not familiar at all with the sometimes bizarre vocabulary of the "tidyverse" (What is a "tibble" or a "kable", for example?) This is clearly not the authors' fault. I would rather suggest to the PeerJ editors to mention expected programming language competences when inviting reviewers. I would most certainly decline any invitation to review R code.

Additional comments

I wonder if the authors have considered making their code available via Binder (see https://github.com/binder-examples/r for an RStudio-based example). The necessity to obtain a Springer API might be a problem for this kind of public execution environment, but perhaps it can be worked around.

·

Basic reporting

no comment

Experimental design

no comment

Validity of the findings

no comment

Additional comments

The work presented takes a timely look at the state of reproducibility in the field of geographic information science. I am not from this field, but passionate enough about reproducibility in science to propose this review. This work is very commendable, and must have taken a fair amount of time and effort. I am broadly supportive of the paper, but cannot recommend publication as is. The aim of this review is thus to help improve the paper, not to criticise the general intrinsic value of the work.
Generally, I found the manuscript very difficult to follow; the structure, and the way the argument is constructed are at odds with what I would have expected.
It is good to provide research questions at the beginning, but let these questions drive and scaffold your narrative. Although your aim is clear, the confusion that arises due to the structure of the article dilutes any take-home message and likely to frighten your readership. Additionally, I can only recommend that the authors ask a native speaker to read and comment on their manuscript; at times, the authors use constructions that don’t really fit, e.g. line 59, one wouldn’t speak of “viewpoint” in this context; one would speak of “points of view” or “opinions".

* Sections 1, 2 and 3.1 seem to belong together; they are introducing the general context of the work, laying the foundations to the narrative. In particular, I don’t see Section 3.1 being part of “materials and methods”.
* Each research question should have its own section, and sections should be named appropriately, not assuming that the reader knows what you are talking about. The type of structure chosen for this paper, interweaving studies in between common sections, only work when studies are related, based on the same methods or assumptions. In this case, by virtue of the work, it adds confusion. Each study/research question could have its own set of sections (material and methods, analyses, etc).

The way this paper reads:
1. Intro
2. Material & Methods
2.a Study 1
2.b Study 2
3. Results
2.a Study 1
2.b Study 2
4. Analyses
2.a Study 1
2.b Study 2
etc

The way I would think makes more sense:
1. Intro
2. Study 1 - RQ1
2.a Material & methods
2.b Analyses
2.c Results
2.d Conclusion
3. Study 2 - RQ2 (or maybe combinations of RQs)
2.a Material & methods
2.b Analyses
2.c Results
2.d Conclusion
4. Summary & Conclusion
etc.

* Section 3.2 contains subsections that are either related to the “investigation”, e.g. Data, Methods, and “informative”, e.g. Summary, and not related to the investigation. These should be distinct.
* Taking the plunge into a whole corpus of papers is valuable, I think, but limiting the analysis to “exploratory text analysis” seems slightly wasteful. I am sure there would be ways for the authors of the papers in the corpus to reference reproducibility-related procedures, etc without using the keywords that the authors of this paper looked at. In other words, it feels that the keywords are rather limited, and not very informative. The more thorough evaluation of the papers that the authors report is, on the other hand, mightily more informative and should be given more space.
* Section 3.4 is a key part of the work, and the authors should spend more time detailing the procedure they followed to assess and rate the publications.
* Why is Table 2 presented before Table 3?

---

## Round 0.2 · accepted · Accept

Dear all,

I'm happy to accept your decision. I've also re-read the manuscript and I think the changes you made address all the comments from both reviewers.

·

Basic reporting

no comment

Experimental design

no comment

Validity of the findings

no comment

Additional comments

The revised manuscript addresses all of my concerns. I particularly appreciated the authors' efforts to make their computational environment available via Binder for easy re-use, and also provide instructions for re-building the Docker image locally. I have tested both methods and could produce the analysis PDF with either one. However, I was not able to generate the corpus-dependent results because I did not manage to get a Springer API key in a reasonable amount of time. The remaining results look visually identical to those provided by the authors. I am not aware of any tool that would let me compare two PDF files (including images) in a more precise way.

·

Basic reporting

n/a

Experimental design

n/a

Validity of the findings

n/a

Additional comments

I am really pleased with the revision of the article. I command the authors on the added description about the reproducibility of the article itself, and further suggestions to authors.

I have noted a couple of glitches:
* line 64, change “viewpoint” to “opinion"
* line 94, add “A Nature Editorial” or “In an editorial in the journal Nature, authors define…"